# Vitamin D and Type 1 Diabetes Risk: A Systematic Review and Meta-Analysis of Genetic Evidence

**DOI:** 10.3390/nu13124260

**Published:** 2021-11-26

**Authors:** Liana Najjar, Joshua Sutherland, Ang Zhou, Elina Hyppönen

**Affiliations:** 1Australian Centre for Precision Health, Unit of Clinical and Health Sciences, University of South Australia, P.O. Box 2471, Adelaide, SA 5001, Australia; najly002@mymail.unisa.edu.au (L.N.); joshua.sutherland@mymail.unisa.edu.au (J.S.); Ang.Zhou@unisa.edu.au (A.Z.); 2South Australian Health and Medical Research Institute, Adelaide, SA 5000, Australia

**Keywords:** diabetes mellitus, type 1, meta-analysis, polymorphism, single nucleotide, vitamin D, 25-hydroxyvitamin D, CYP2R1

## Abstract

Several observational studies have examined vitamin D pathway polymorphisms and their association with type 1 diabetes (T1D) susceptibility, with inconclusive results. We aimed to perform a systematic review and meta-analysis assessing associations between selected variants affecting 25-hydroxyvitamin D [25(OH)D] and T1D risk. We conducted a systematic search of Medline, Embase, Web of Science and OpenGWAS updated in April 2021. The following keywords “vitamin D” and/or “single nucleotide polymorphisms (SNPs)” and “T1D” were selected to identify relevant articles. Seven SNPs (or their proxies) in six genes were analysed: *CYP2R1* rs10741657, *CYP2R1* (low frequency) rs117913124, *DHCR7/NADSYN1* rs12785878, *GC* rs3755967, *CYP24A1* rs17216707, *AMDHD1* rs10745742 and *SEC23A* rs8018720. Seven case-control and three cohort studies were eligible for quantitative synthesis (*n* = 10). Meta-analysis results suggested no association with T1D (range of pooled ORs for all SNPs: 0.97–1.02; *p* > 0.01). Heterogeneity was found in *DHCR7/NADSYN1* rs12785878 (I^2^: 64.8%, *p* = 0.02). Sensitivity analysis showed exclusion of any single study did not alter the overall pooled effect. No association with T1D was observed among a Caucasian subgroup. In conclusion, the evidence from the meta-analysis indicates a null association between selected variants affecting serum 25(OH)D concentrations and T1D.

## 1. Introduction

Type 1 diabetes (T1D) is a chronic autoimmune disease, resulting from autoimmune degradation of pancreatic ß-cells leading to the inability to produce and/or use insulin [1]. T1D patients carry a genetic susceptibility to autoimmune disease development, with first-degree relatives of those affected also carrying an increased risk of developing the disease [2,3]. Undiagnosed or untreated T1D can result in hyperglycaemia, increasing the risk of developing microvascular and macrovascular injuries/health complications, such as nephropathy, ischemic heart disease and stroke [4]. Estimates of those with T1D below age 20 had risen to over a million in 2017, with evidence of increasing incidence worldwide [5]. Presently, there are no established treatments identified for the prevention of T1D and the search for genetic and environmental triggers remains ongoing.

Emerging evidence suggests low vitamin D status may play a role in T1Dpredisposition. Vitamin D is a steroid prohormone, with nutrition status approximated via serum 25-hydroxyvitamin D [25(OH)D] concentrations [6]. Notably, 25(OH)D deficiency is strongly associated with skeletal pathology, however, in the advent of vitamin D receptors being discovered throughout the body, there now is a greater acknowledgment of broader disorders associated with deficiency, including autoimmune issues, such as T1D and multiple sclerosis [7,8]. Recent evidence indicates an important role for active vitamin D [1,25(OH)2D] in immune regulation [9]. Mechanistic explanations for 1,25(OH)2D include immunomodulatory action leading to cytokine regulation, reducing the likelihood of destruction of pancreatic ß-cells [10]. Another potential mechanism is through direct protection of pancreatic ß-cells, serving to preserve barrier exclusion of pathogens, likely significant in the prevention of autoimmune disorders [11]. Such mechanistic insight has underpinned novel immune-modulatory concepts for the prevention of T1D.

Association between serum 25(OH)D concentrations and T1D risk is supported by evidence from in vitro and animal experiments [12,13,14], as well as human observational studies [15,16,17,18] and ecological correlation [19]. In animal studies, oral administration of the activated form of vitamin D was found to protect nonobese diabetic mice from T1D [12,13,14], while human observational studies have shown reduced levels of serum 25(OH)D are associated with increased risk of T1D [15,17]. In the aetiology of T1D observational studies have also shown support of vitamin D supplementation in being inversely associated with T1D [16,18,20]. Animal experimental data, therefore, indicate low 25(OH)D concentrations may be involved in T1D predisposition, however, a causal role of impaired vitamin D metabolism in the aetiology of T1D in humans is yet to be implicated, and stronger forms of evidence—less effected by confounding or reverse causation—are required.

Using selected vitamin D related genetic variants, it is possible in a genetic epidemiological setting to establish evidence of an etiological role of 25(OH)D in T1D pathophysiology. Since 25(OH)D synthesis is regulated by genes, single nucleotide polymorphisms (SNPs) may alter the bioavailability and target effects of vitamin D metabolites. Large-scale genome-wide association studies (GWAS) have identified several SNPs from genes influencing 25(OH)D levels; *CYP2R1*, *DHCR7/NADSY1*, *GC*, *CYP24A1*, *AMDHD1* and *SEC23A*, which have been used as genetic instrumental variants in this study [21,22].

As individual studies may not have enough statistical power to identify an association between selected genetic variants affecting serum 25(OH)D concentrations and T1D, a meta-analysis is a useful statistical tool to pool data from published studies, where increasing the statistical power can give more accurate estimates of effect sizes. In this study, we perform a systematic review and meta-analysis of all existing studies reporting an association between selected 25(OH)D related genetic variants (exposure) and T1D risk (outcome) in humans (population). This topic provides a further scientific understanding of T1D pathophysiology and the potentiality of preventing T1D through increases in 25(OH)D concentrations.

## 2. Materials and Methods

This systematic review and meta-analysis followed the Preferred Reporting Items for Systematic Reviews and Meta-Analyses (PRISMA) guidelines [23]. Registration: PROSPERO (ID CRD42021224844), https://www.crd.york.ac.uk/prospero/ (accessed on 10 January 2021).

### 2.1. Search Strategy

A search was conducted in four databases: Ovid Medline (1964-present), Ovid Embase (1947-present), Web of Science (1975-present), IEU OpenGWAS (2020-present) from inception to April 2021. The primary search terms were as follows: humans, single nucleotide polymorphism, genetic variation, type 1 diabetes mellitus and vitamin D. The selection of articles in Medline and Web of Science was performed using Medical Subject Headings (MeSH) to define these descriptors. The selection of articles in Embase was performed using Emtree (Embase subject headings) to define these descriptors. Boolean operators (e.g., OR, AND, NOT) were also combined with keywords and subject headings. An initial pilot search was undertaken to improve inclusion clarity of study inclusion and exclusion, improving accuracy and consistency. The strategy was developed by one reviewer (L.N.) and proofread for syntax, spelling and overall structure by two reviewers (E.H. and J.S.). As part of the development process, we used two relevant, existing studies [24,25] for validation purposes, testing if our search strategy could identify them. The set of search terms was slightly modified between databases due to different system procedural limitations, however, the overall approach remained as consistent as possible across each database. The selection of studies through OpenGWAS, as well as the UK Biobank, was prepared using R 4.0.2 software, conducting an SNP-based search for the selected genetic variants and their proxies (r^2^ > 0.8), locating any additional studies fitting the inclusion criteria. Full search strategies are presented in Appendix A.

### 2.2. Inclusion and Exclusion Criteria

Studies testing exposure of selected genetic variants or their proxies with r^2^ > 0.8 influencing 25(OH)D pathways for association with T1D status and 25(OH)D concentrations, were of interest. Eligible studies met the population, exposure, outcome (PEO) approach [26] as follows:1.Population: human of any gender and age, race and geographical distribution.2.Exposure: a biological approach to the selection of genetic variants was used, including variants having a biological link to the exposure. Seven vitamin D related SNPs were selected: *CYP2R1* (common variant) rs10741657, *CYP2R1* (rare variant/low frequency) rs117913124, *DHCR7/NADSYN1* rs12785878, *GC* rs3755967, *CYP24A1* rs17216707, *AMDHD1* rs10745742, *SEC23A* rs8018720. Of these selected SNPs, six are common variants identified based on the results of a recent GWAS for 25(OH)D concentration [21] and one is a low-frequency synonymous coding variant seen with a much larger effect on 25(OH)D concentration [22]. Strong genome-wide associations with 25(OH)D were found in genes located upstream (*DHCR7/NADSY1* and both *CYP2R1* variants), and two downstream (*CYP24A1* and *GC*) of the 25(OH)D metabolite biochemical pathway. Two genes outside the vitamin D metabolism pathway (*AMDHD1* and *SEC23A*) were also found to be significant variants and hence were included. 25(OH)D related proxies not directly present in the recent GWAS were also included if found in high linkage disequilibrium (r^2^ > 0.8) using the Ldproxy function in LD link (https://ldlink.nci.nih.gov, accessed on 12 April 2021).3.Outcome: the primary outcome measure, T1D, was defined by the World Health Organization criteria: diabetes symptoms (polyuria, polydipsia and insulin deficiency), accompanied by exogenous insulin usage once T1D had been diagnosed [27]. T1D could be self-reported or doctor-diagnosed when confirming cases.4.Study design: peer-reviewed genetic association, cohort, cross-section, or case-control observational studies and Mendelian randomization (MR) studies, as well as clinical trials and unpublished cohort studies.5.A sample size of at least 50 cases and 50 controls were mandatory for sufficient data extraction. Where there were multiple publications from the same study population, the most recent highest quality results with the largest sample size were used.6.The publication reported genotype distribution in both cases and controls in order to estimate an odds ratio (OR) with a 95% confidence interval (CI).

The following exclusion criteria were also used:1.Conference papers.2.Other types of diabetes.

No language, publication status, or publication date limitations were imposed.

### 2.3. Study Selection and Data Extraction

Literature was searched in duplicate independently by two authors (L.N. and J.S.), and approved by a third author (E.H.). After excluding duplicates, article selection was carried out in two passes. In the first pass, title and abstract screening occurred for the selection of relevant papers meeting the eligibility criteria. In the second pass, proposed articles from the first pass were screened in full text for compliance with inclusion criteria. To ensure literature saturation, reference lists of obtained studies from original database searches were manually scanned for potential unidentified additional studies by one author (L.N.), with eligibility confirmed by a second author (J.S.). Furthermore, OpenGWAS was used to identify unpublished studies, locating one FinnGen cohort study sharing summary-level data fitting the search parameters. Datasets were also identified in the UK Biobank, a large-scale prospective cohort study.

Data were extracted independently by two authors (L.N. and J.S.) using a predetermined data extraction template. The following data were extracted from the articles included in this systematic review: first author; region/demographic information; publication year; study design characteristics; participant characteristics, including gender and ethnicity if reported; the number of cases and controls studied; mean age (or range) at the onset of T1D in cases; outcome measure, diagnostic criteria of T1D; mean age (or range) of the control group; how the controls were selected; genotyping methods, genotype distribution, and allele frequency in cases and controls; all reported patient outcome measures; key findings; protocol availability and funding sources. Corresponding authors were contacted by e-mail for missing or unreported data a maximum of three attempts, to avoid any assumptions made from unclear information. All disagreements were resolved by consensus, or with the input of a third author (E.H. or A.Z.).

### 2.4. Statistical Analysis

All mentioned statistical analyses were performed with STATA 16.0 software (Stata Corporation, College Station, TX, USA) and R 4.0.2 software by two authors (L.N. and A.Z.). For each variant, OR per vitamin D-increasing allele was extracted from individual studies for the meta-analysis, as per the SUNLIGHT consortium [21]. If a study did not contain the selected vitamin D variant, the result of its proxy (r^2^ > 0.8) was extracted and used to estimate the related effect. In studies where the OR per vitamin-D-increasing allele was not reported, we estimated the allelic effect from the contingency table of T1D distribution by SNP genotypes, where OR was computed by dividing the odds of T1D in the heterozygotes (i.e., with 1 25-hydroxyvitamin D increasing allele) by that in the homozygotes (i.e., with 0 25-hydroxyvitamin D increasing allele). Meta-analysis was performed using the random-effects model (REM, restricted maximum likelihood method) [28]. Heterogeneity between studies was assessed using Cochran’s Q test the I^2^ statistic, with heterogeneity considered to be substantial if the *p*-value for the Cochran’s Q test < 0.05 or I^2^ > 50%. All *p*-values were for two-tailed tests, and <0.05 was considered statistically significant.

We conducted sensitivity analyses by removing a single study at a time, evaluating the integrity of the results. Subgroup analysis was performed by restricting the sample to the Caucasian population, to examine the possible effects of population stratification. Initial protocol pre-specified plan for further MR analyses, which were not conducted as it was considered redundant given clear results.

### 2.5. Risk of Bias and Credibility of the Evidence Assessment

The methodological quality of eligible studies was evaluated using Critical Appraisal Skills Program tools for cohort and case-control studies [29]. Two authors (L.N. and J.S.) independently completed risk of bias assessment and recorded supporting justification and information for each domain to optimise the tool’s value (met; partially met; not met; unclear). The domains were: Are the results of the study valid? Were the cases recruited in an acceptable way? Was the exposure accurately measured to minimise bias? Have authors taken account of the potential confounding factors in the design and/or in their analysis? How precise are the results? (size of confidence intervals). Results were compared by categorising each study for study quality (risk of bias) judgement (low, some concerns, high). Articles were judged as ‘low’ when five or more domains were met. Conversely, articles were judged as ‘high’ when three or more domains were unmet. Disagreements were resolved by a third author (E.H.). Outcome reporting bias was assessed by comparing outcomes specified in protocols, with outcomes reported in corresponding publications. Where protocols were not available, outcomes specified in the methods and results sections of publications were compared.

Two reviewers assessed the risk of bias due to missing results in a synthesis (L.N. and A.Z.). Potential publication bias was assessed by examining for asymmetry using Begg’s funnel plot for each SNP [30]. If publication bias was present, the plot would be asymmetric, indicating a deficiency in publications with negative results. No further formal assessment of publication bias, such as Egger’s test was performed, due to insufficient studies [31].

## 3. Results

### 3.1. Study Selection

Initially, 290 potential studies were identified from the search. Figure 1 shows a flowchart of the study selection process based on the PRISMA statement [23]. After the initial pass, 58 were excluded as duplicates. 212 were excluded after reading the title and abstract because of evident irrelevance. In the second pass, the full text of the 20 studies selected in the first pass were read and 10 studies were excluded for not meeting the search criteria. Two articles were excluded because they did not provide sufficient data for the calculation of Ors with 95% CI [32,33]. Three papers were excluded because they were family-based [34,35,36]. Two papers were excluded as they assessed associations between polymorphisms not in linkage disequilibrium with the selected variants [37,38]. Two papers did not investigate the association between the selected variants and T1D, investigating a different outcome [39,40]. Only one study was excluded due to using the same sample population [24]. Therefore, 10 studies were included in this systematic review.

### 3.2. Characteristics of Included Studies

The summary characteristics of included studies are shown in Table 1. The studies were published between 1999 to 2021, conducted in different geographical locations. Of the 10 included studies, seven were case-control studies [25,40,41,42,43,44], and three had a cohort design [45,46,47]. Most studies focused on T1D in childhood, as indicated by the mean age of onset in cases. Appropriate genotyping methods and diagnostic criteria were used in all included studies. Of the studies selected, six studies [40,41,42,43,47] fulfilled the WHO diagnostic criteria for T1D, while the majority of the remaining studies [25,44,45,46,48] indirectly captured criteria by description from multiple case sample populations. Polymerase chain reaction-restriction fragment length polymorphism (PCR-RFLP) was used by half the included studies as the genotyping method.

Similarly, none of the eligible studies endeavoured to control for vitamin D dietary intake through infancy and/or childhood, a known risk factor of T1D. However, when study quality was assessed, all included studies presented with a low risk of bias using the CASP tools, with no deviation from the Hardy-Weinberg equilibrium in controls reported in all case-control studies, and only some studies presenting with one item partially unmet (Appendix A).

Statistical methods to control confounding varied between studies. Most studies adjusted for different potential confounding factors, such as age, sex, genotype batch, geographical origin and BMI (see Table 1). Two remaining papers were matched case-control studies to control for known potential confounding variables. Hussein et al. [41], matched by age and ethnic origin, while Mahmoud et al. [42] matched by gender. Six studies [40,43,44,45,46,47] did not report OR results directly, and some, but not all, of the studies, generated adjusted ORs.

### 3.3. Findings from the Meta-Analysis

All specified polymorphisms (namely rs10741657 G/A (*CYP2R1*), rs117913124 A/G (*CYP2R1* low frequency), rs12785878 G/T (*DHCR7/NADSYN1*), rs3755967 T/C (*GC*), rs17216707 C/T (*CYP24A1*), rs10745742 C/T (*AMDHD1*), rs8018720 C/G (*SEC23A*) were reported in three or more studies and taken forward to the meta-analyses. Associations between the SNPs and T1D, using individual and pooled OR estimates, are displayed in Figure 2 and Appendix A.

For rs10741657 G/A (*CYP2R1*), the reported ORs ranged from 0.46 to 1.11 (Figure 2). The random-effects pooled OR was 0.97 (95% CI 0.93, 1.02; *p* = 0.01) with little heterogeneity among the studies (I^2^ = 25.1%). For rs117913124 A/G (*CYP2R1* low frequency), the ORs ranged from 1.00 to 1.07 (Figure 2) with a pooled OR of 1.02 (95% CI 0.94, 1.11; *p* = 0.78; I = 0.0%). For rs12785878 G/T (*DHCR7*/*NADSYN1*), the ORs ranged from 0.78 to 1.06 (Figure 2), with a pooled OR of 0.99 (95% CI 0.92, 1.07; *p* = 0.02). There was evidence of moderate between-study heterogeneity (I^2^ = 64.8%). For rs3755967 T/C (GC), the OR ranged from 0.99 to 1.53 (Figure 2), with a pooled OR of 1.02 and no sign of heterogeneity (95% CI 0.99, 1.06; *p* = 0.97; I = 0.0%). In the evaluation for publication bias, asymmetry in Begg’s funnel plot was observed for *GC* rs3755967 (Appendix A). For rs17216707 C/T (*CYP24A1*), the OR ranged from 0.96 to 1.03 (Figure 2). The random-effects model pooled OR was 1.00 (95% CI 0.95, 1.04, *p* = 0.37), with little indication of heterogeneity (I^2^ = 18.0%). For rs10745742 C/T (*AMDHD1*), the OR ranged from 1.00 to 1.02 (Figure 2) with a pooled OR of 1.00 (95% CI 0.97, 1.04; *p* = 0.90). Again, there was no sign of heterogeneity (I^2^ = 0.0%). For rs8018720 C/G (*SEC23A*), the OR ranged from 0.97 to 1.05 (Figure 2). The REM yielded a pooled OR of 1.01 (95% CI 0.95, 1.07, *p* = 0.19) with little heterogeneity among the studies (I^2^ = 42.8%). In view of these individual estimates, under the studied models no statistically significant associations between any of the seven SNPs alone (or their proxies) and T1D were found. Other than in rs3755967 (*GC*), no other asymmetry in Begg’s funnel plot was observed. No outcome reporting bias was detected in any of the studies.

Furthermore, a sensitivity analysis was also performed to assess the influence of each study using the leave-one-out method. The pooled ORs were not changed materially and remained not significant, indicating good stability of results (range of pooled OR: 0.97–1.02). A subgroup analysis performed on the Caucasian population found no manifestations of association, with no major changes in primary outcomes (Appendix A). Analyses showed all seven selected polymorphisms (or their proxies) were not associated with T1D risk under the studied models (range of pooled OR: 0.98–1.02).

## 4. Discussion

### 4.1. Main Findings

Our extensive systematic review and meta-analysis did not provide support for an association between 25(OH)D related variants and T1D. Our review identified 10 studies for inclusion, which were all relatively high quality, presenting only minor systematic flaws in methodology. However, evidence from published studies was inconsistent, and for most polymorphisms, only a handful of studies were found. Many of the studies were small, limiting the statistical power of each meta-analysis, and preventing robust sensitivity analyses to evaluate associations by possible sources of heterogeneity, such as geographic location, and ancestry.

To the best of our knowledge, our study is the largest and most comprehensive systematic review and meta-analysis on the topic. The largest of the previous studies was a recently published MR study [45], which also provided a null finding, and from which raw data were included in this study. We conducted leave-one-out analyses, which suggested limited impact by any single study, alleviating concerns for bias caused by the inclusion of smaller or early studies. Furthermore, ethnicity is believed to have a major role in vitamin D synthesis (and possibly metabolism), however, subgroup analysis on Caucasian participants also provided no evidence for an association between the selected 25(OH)D related genetic variants and T1D.

From publications included in our review, those studies which found evidence for an association with T1D risk, tended to be comparatively small, while the association could not be confirmed in the large genetic databases. For example, Ramos-Lopez et al. [40] found an association of the *CYP2R1* common variant polymorphisms with T1D in 578 German participants, providing early support for the causal role of 25(OH)D in the pathogenesis of T1D. Hussein et al. [41] also found an association in an Egyptian sample (*n* cases = 120) between the *CYP2R1* common variant with risk of T1D. Smaller study over-estimates of effect can yield asymmetric funnel plots that can be explained by a restrictive study population [49]. However, the two smaller studies reporting an association included in this paper, had a matched case-control design, suggesting a possibility they were more carefully designed than the larger database based studies. For example, case ascertainment in the database studies typically had diagnoses confirmed by self-report or hospitalisation. Furthermore, despite including participants from diverse ethnic groups, Hussein and colleagues, had an ethnicity-matched control sample [41]. In contrast, recent larger studies in the European population including between 350 and 9358 cases [25,45,46], as well as our analyses including 3221 cases (387,397 controls) from the UK Biobank, did not find evidence for an association between any of the selected genetic variants and T1D. While we did not find evidence for publication bias, there was possible asymmetry in Begg’s funnel plot for *GC* rs3755967 (Appendix A). However, its interpretation should be taken as merely an evaluation of whether smaller studies gave different results to larger studies, as further formal testing for publication bias would have been largely underpowered due to the limited number of studies.

High heterogeneity was found in the meta-analysis *DHCR7/NADSYN1* rs12785878 polymorphism, (I^2^ = 64.8%), which was unanticipated given the studies included in the analyses of this variant were all of European ancestry, with adjustments for confounding factors. However, DHCR7 affects skin synthesis of vitamin D following exposure to UVB radiation from the sunlight and may be particularly sensitive to subtle variations in population structure. Variants affecting vitamin D metabolism have been shown to display population-specific patterns in frequency [50], and are believed to have contributed to adaptations during the evolutionary history which has allowed individuals to avoid severe vitamin D deficiency [51]. This has been seen in earlier vitamin D related genetic meta-analyses, which have allowed for the examination of population stratification. Notably, a large meta-analysis found the *Bsml* polymorphism in the vitamin D-receptor gene was only associated with T1D in those with Asian ancestry [52]. Differing environmental factors, such as geographical differences in diet and sun exposure, may also play a role to aggravate or compensate susceptibility conferred by variants in these genes [53].

### 4.2. Considerations of Alternative Explanation for Observed Results

Vitamin D status is mainly determined by lifestyle factors, such as exposure to sunlight, dietary supplementation and intake, as well as personal characteristics including obesity and age. Indeed, common genetic variants typically have modest effects, and they only account for a small amount of the variation in 25(OH)D levels [54]. Therefore, even if variation in 25(OH)D concentrations is important for T1D, but only at the very extreme (such as clinical deficiency), this type of genetic instrument may not be able to pick up an association, especially if most of the population investigated has relatively normal concentrations. The influence of genetic variations may also be affected by interactions with other genes and by environmental factors.

Given the limited number of studies, we were unable to assess ethnic differences in the association between 25(OH)D variants and T1D. Given ethnicity may affect the function and expression of vitamin D related genes [50,52,53], it is possible that we may have missed associations that are only seen in a particular population group.

### 4.3. Strengths and Limitations

Our study benefits from the systematic way in which results have been summarised, our comprehensive search strategy and the inclusion of grey literature. The design captures lifetime differences in 25(OH)D levels, rather than a single vitamin D measurement. Our study also has some limitations requiring consideration. Despite including information from the largest available databases to supplement all published data, available information remains limited. The relatively small number of included studies prevented us from undertaking analysis to examine the associations in diverse ethnic groups or to account for other population characteristics. There was little to no information from populations that were vitamin D deficient. For example, the FinnGen study in Finland commenced in 2017, after the National Nutrition Council had launched the national food fortification of vitamin D (2002) [55]. Therefore, we are unable to exclude weak associations or associations that are only relevant in the context of very low 25(OH)D concentrations. We did not have access to individual level data for most of the studies, therefore, adjustments strategies could not be harmonised. Results could be limited by the absence of dietary information for all study participants, as studies have shown an association between vitamin D genes can vary due to diet, or even past sun exposure [56]. Furthermore, evidence participants of the UK Biobank are not representative of the UK population, having a healthy volunteer selection bias [57]. Thus, we are only able to investigate for a causal effect within the constraints of each study, which may have contributed to the null finding.

### 4.4. Guidelines for Future Research

Investigating for smaller causal effects may be important for public health, due to a high prevalence of low 25(OH)D concentrations in many populations. Findings need to be elucidated by conducting larger scale epidemiological investigations, exploiting the potential for vitamin D related genetic variants as a risk factor for T1D, to confirm or refute the study findings. Furthermore, said studies will need to investigate the role of 25(OH)D related genetic variants in the context of clinical deficiency, where even subtle increases in concentrations may help, providing a more comprehensive understanding of the association between variants affecting serum 25(OH)D concentration and T1D.

## 5. Conclusions

Results from this meta-analysis showed no large effect of a genetically determined reduction in 25(OH)D concentrations by selected polymorphisms on T1D risk, despite the strong association seen in some observational studies. Although the hypothesis that a different SNP distribution from vitamin D related genes is associated with T1D was not confirmed by this study, small effects cannot be discounted. To make conclusive estimates in complex diseases, such as T1D, further characterization of complex interactions between genetic and environmental factors, like the included variants affecting serum 25(OH)D concentrations, need to be considered.

## Figures and Tables

**Figure 1 nutrients-13-04260-f001:**
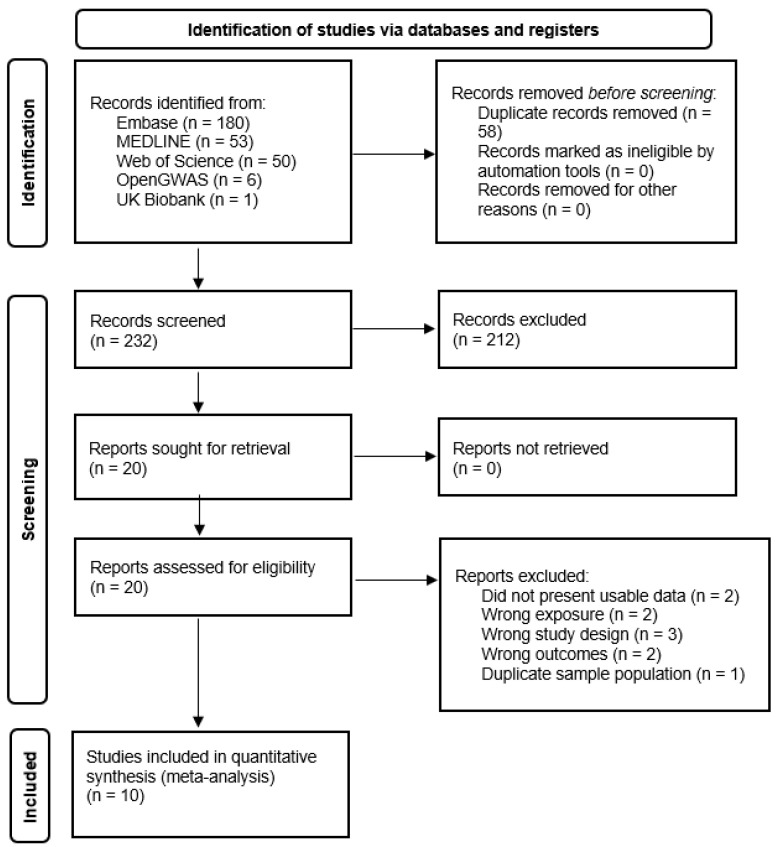
Flowchart illustrating the literature search and study selection.

**Figure 2 nutrients-13-04260-f002:**
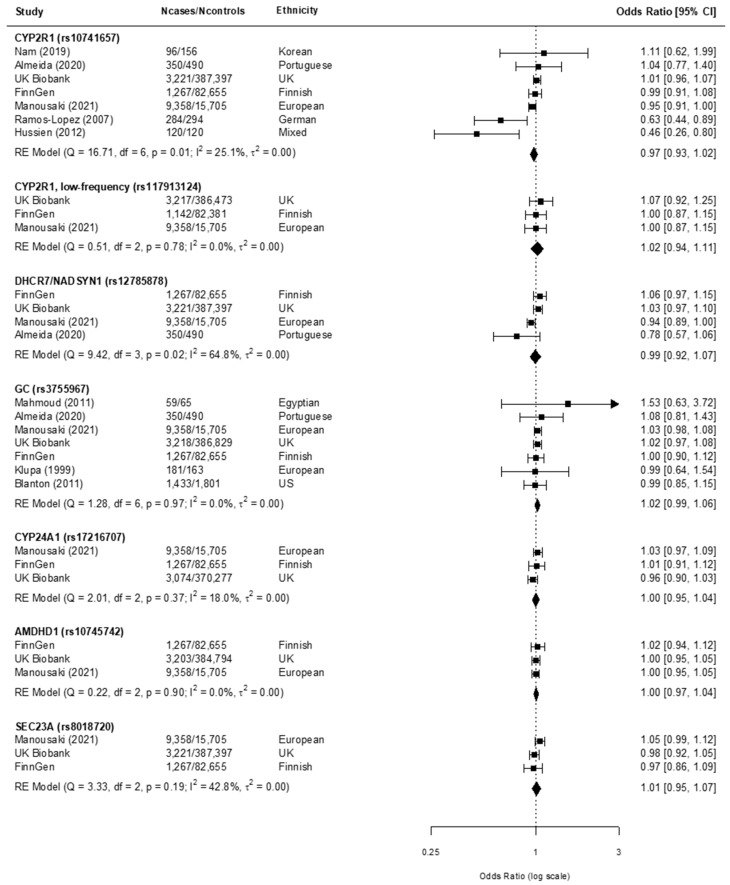
Meta-analysis for the association between selected genetic variants affecting serum 25-hydroxyvitamin concentrations and type 1 diabetes with the random effects model (variants coded by 25-hydroxyvitamin D increasing alleles). Squares represent the individual odds ratio estimate. Diamonds show the pooled effect. Horizontal bars represent the 95% confidence intervals.

**Table 1 nutrients-13-04260-t001:** Characteristics of observational studies evaluating the association between vitamin D genetic variants and type 1 diabetes included in the meta-analysis.

Study Details					Participant Characteristics			Polymorphism Details					Findings
Author; Year	Country	Study Design	Ethnicity	*n* Cases/*n* Controls	Mean Age of Cases/Controls (Year)	Mean Age of Onset in Cases (Years)	T1D Diagnostic Criteria	Genotyping	Adjusted Factors	Gene	Variant	EA ^a^	Relevant Key Findings
Manousaki et al., 2021 [45]	Canada, United Kingdom, United States	Cohort	European	9358/15,705	NI	NI	Multiple criteria	PCR-RFLP	Age, sex, season of 25OHD measurement, genotype batch, genotype array, assessment centre (proxy for latitude)	*CYP2R1* *CYP2R1 (low frequency)* *DHCR7/NADSYN1* *GC* *CYP24A1* *AMDHDI* *SEC23A*	*rs10741657* *rs117913124* *rs12785878* *rs3755967* ^b^ *rs17216707* *rs10745742* *rs8018720*	A G T C ^c^ T T G	No association of individual SNPs with T1D.
Almeida et al., 2020 [25]	Portugal	Case-control	Caucasian Portuguese	350/490	29.0/32.2	17.2	Classic clinical presentation ^d^	PCR-RFLP	Age at bleed, sex, BMI, month of bleed, geographical region	*CYP2R1* *DHCR7/NADSYN1* *GC*	*rs10741657* *rs12785878* *rs3755967* ^b^	A T C ^c^	No association of individual SNP with T1D.
Nam et al., 2019 [44]	Korea	Case-control	Korean	96/156	14.7/14.0	NI	Classic clinical presentation ^d^	PCR	25OHD and 1α,25(OH)2D levels. (25OHD measurement obtained in same season)	*CYP2R1*	*rs10741657*	A	No association of individual SNP with T1D.
Hussein et al., 2012 [41]	Egypt	Matched case-control	Egyptian	120/120	11.7/11.1	NI	WHO and ADA	PCR-RFLP	Nil	*CYP2R1*	*rs10741657*	A	An association of GG genotype of CYP2R1 polymorphism (coded by 25(OH)D decreasing alleles) with risk of T1D in Egyptian children [OR = 2.6, 95% CI = 1.1–6.1, *p* = 0.03]. A synergistic effect of multiple risk alleles between GG genotype of CYP2R1 and CC genotype of CYP27B1 and T1D risk found.
Mahmoud et al., 2011 [42]	Egypt	Matched case-control	Egyptian	59/65	13/>24	7.5	WHO	PCR-RFLP	Nil	*GC*	*rs3755967* ^b^	C ^c^	No association between VDBP polymorphisms with T1D.
Blanton et al., 2011 [48]	United States	Case-control	American	1705/2033	NI	12.9	Classic clinical presentation ^d^	TaqMan PCR Assays	Sex, onset of T1D, HLA risk	*GC*	*rs3755967* ^b^	C ^c^	No association between VDBP polymorphisms with T1D detected. An association of the phenotype of lower VDBP levels with T1D.
Ramos-Lopez et al., 2007 [40]	Germany	Case-control	German	284/294	NI	11.5	WHO	PCR-RFLP	25(OH)D3 levels	*CYP2R1*	*rs10741657*	A	An association of the ‘G’ allele of CYP2R1 common variant polymorphisms (coded by 25(OH)D decreasing alleles) with T1D risk.
Klupa et al., 1999 [43]	United States	Case-control	European	181/163	36.2/52.55	10.9	WHO	PCR	Nil; sensitivity confirmed via stratification by obesity and age at examination	*GC*	*rs3755967* ^b^	C ^c^	No association of individual SNP with T1D.
FinnGen [46]	Finland	Cohort	Finnish	1143–1267/82,381–82,655	NI	NI	Strict definition (Minimal/absent insulin production by pancreas)	Illumina and Affymetrix Chip Arrays	Sex, age, 10 PCs, genotyping batch	*CYP2R1* *CYP2R1 (low frequency)* *DHCR7/NADSYN1* *GC* *CYP24A1* *AMDHDI* *SEC23A*	*rs10741657* *rs117913124* ^b^ *rs12785878* *rs3755967* *rs17216707* *rs10745742* *rs8018720*	A G ^c^ T C T T G	NI
UK Biobank [47]	United Kingdom	Cohort	Caucasian British	3074–3221/370,277–387,397	NI	NI	WHO	UK Biobank Axiom Array	Age, sex, birth location, assessment centre, SNP array, pc1-pc40, account for relatedness	*CYP2R1* *CYP2R1 (low frequency)* *DHCR7/NADSYN1* *GC* *CYP24A1* *AMDHDI* *SEC23A*	*rs10741657* *rs117913124* *rs12785878* *rs3755967* *rs17216707* *rs10745742* *rs8018720*	A G T C T T G	NI

Abbreviation: 25(OH)D, 25-Hydroxyvitamin D; *n*, number; T1D, type 1 diabetes; NI, not informed; ADA, American Diabetes Association; WHO, World Health Organization; PCR, polymerase chain reactions; PCR-RFLP, polymerase chain reaction-restricted fragment length polymorphism; SNP, single nucleotide polymorphism.; Vit D, vitamin D; EA, effect allele; OR, odds ratio; VDBP, vitamin D binding protein. ^a^ Each effect allele represents the 25(OH)D concentration increasing allele, as defined by Sunlight Consortium [21]. ^b^ Identified using LDproxy, coded by 25(OH)D concentration decreasing alleles (see methods) ^c^ Effect allele direction reversed based on 25(OH)D concentration increasing, as defined by Sunlight Consortium (see methods) [21]. ^d^ Low/undetectable serum C-peptide and presence of 1+ pancreatic autoantibodies.

## Data Availability

Not applicable, due to being systematic review with meta-analyses. All data is available in primary studies.

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
