# Peer review of "Vitamin D and Type 1 Diabetes Risk: A Systematic Review and Meta-Analysis of Genetic Evidence"

_nutrients, 2021, doi:10.3390/nu13124260_

Round 1
Reviewer 1 Report
The authors have done a nice job studying the link between vitamin D and DM type 1.
There are however some comments.
- In the exposure, 7 variants were selected by the authors. Considering the limited number of the included articles, wouldnt it make more sense if they had included all the variants?
- In the discussion, where they discuss the other explanation for their results, what about the effect of ethnicity. We know certain vit D variants and polymorphisms are more common in certain ethnicities, couldnt this be a cause?
- Have you tried doing the analysis while taking into account the ethnicities, I understand that there are limited number of articles and maybe this is not possible.
- In the results and methods section, there is no explanation of how the quality of the articles were checked.
Author Response
The authors have done a nice job studying the link between vitamin D and DM type 1.
Thank you for this positive assessment of our paper.
There are however some comments.
- In the exposure, 7 variants were selected by the authors. Considering the limited number of the included articles, wouldn’t it make more sense if they had included all the variants?
We selected candidate variants which have confirmed/replicated association with 25(OH)D and a clear role in the metabolic vitamin D pathway. We agree with the reviewer that an alternative approach would have been to select all variants which have been suggested to be associated with 25(OH)D concentrations in genome wide association studies (up to 143 loci). “However, we chose not to do this as for most of these variants the role in vitamin D metabolism may be secondary and are likely to be pleiotropic. For example, the 143 loci includes variants in genes such as APOE (dementia, CVD, infection), PDILT (blood pressure), GCKR (serum calcium level, lipids, type 2 diabetes, fatty liver disease, obesity etc.), and CPS1(renal function), among others. Consequently, inclusion of those additional variants would make the assessment of SNP-T1D associations more prone to bias and be potentially misleading. It should also be noted that while we only included seven SNPs instead of the 143, these SNPs captured more than half of the variation explained (2.8% vs. 4.9%), showing that any remaining individual variants would have had only a weak association with 25(OH)D and that we were clearly able to capture the strongest influences.
- In the discussion, where they discuss the other explanation for their results, what about the effect of ethnicity. We know certain vit D variants and polymorphisms are more common in certain ethnicities, couldn’t this be a cause?
We thank the reviewer for this comment, and have now added a related note to the discussion. (lines 421-425)
- Have you tried doing the analysis while taking into account the ethnicities, I understand that there are limited number of articles and maybe this is not possible.
Further exploration of the ethnic differences in the association between genetic vitamin D related genetic variants and type 1 diabetes would be certainly interesting. However, as the reviewer suspects due to the limited number of available studies, further explorations are not currently possible.
- In the results and methods section, there is no explanation of how the quality of the articles were checked.
We thank the reviewer for this insight, and have made relevant adjustments to the results and methods section to reflect this note (lines 233-239, 286).
Reviewer 2 Report
In this paper, Najjar et al. performed a systematic review and meta-analysis of all existing studies (10 included) reporting association between selected 25(OH)D related genetic variants and type 1 diabetes risk in humans. Authors did not find support for an association between 25(OH)D related variants and type 1 diabetes risk.
This is an interesting topic highlight further scientific understanding of type 1 diabetes pathophysiology and the potentiality of preventing type 1 diabetes through increases in 25(OH)D concentrations. This is a methodologically well conducted systematic review and meta-analysis
Only minor suggestions:
Authors could abbreviate type 1 diabetes in T1D
261: juvenile is an obsolete term
Author Response
In this paper, Najjar et al. performed a systematic review and meta-analysis of all existing studies (10 included) reporting association between selected 25(OH)D related genetic variants and type 1 diabetes risk in humans. Authors did not find support for an association between 25(OH)D related variants and type 1 diabetes risk. This is an interesting topic highlight further scientific understanding of type 1 diabetes pathophysiology and the potentiality of preventing type 1 diabetes through increases in 25(OH)D concentrations. This is a methodologically well conducted systematic review and meta-analysis.
We thank the reviewer for the positive assessment of our paper.
Only minor suggestions:
- Authors could abbreviate type 1 diabetes in T1D
Now done.
- 261: juvenile is an obsolete term
Now revised.